# Sodium–Glucose Cotransporter 2 Inhibitors in Patients with Diabetes and Coronary Artery Disease: Translating the Benefits of the Molecular Mechanisms of Gliflozins into Clinical Practice

**DOI:** 10.3390/ijms24098099

**Published:** 2023-04-30

**Authors:** Arturo Cesaro, Vincenzo Acerbo, Erica Vetrano, Giovanni Signore, Gianmaria Scherillo, Francesco Paolo Rotolo, Gianantonio De Michele, Francesco Scialla, Giuseppe Raucci, Domenico Panico, Felice Gragnano, Elisabetta Moscarella, Raffaele Galiero, Alfredo Caturano, Roberto Ruggiero, Ferdinando Carlo Sasso, Paolo Calabrò

**Affiliations:** 1Department of Translational Medical Sciences, University of Campania “Luigi Vanvitelli”, I-80131 Naples, Italy; 2Division of Clinical Cardiology, A.O.R.N. “Sant’Anna e San Sebastiano”, I-81100 Caserta, Italy; 3Department of Advanced Medical and Surgical Sciences, University of Campania Luigi Vanvitelli, Piazza Luigi Miraglia 2, I-80138 Naples, Italyferdinandocarlo.sasso@unicampania.it (F.C.S.)

**Keywords:** SGLT2 inhibitors (SGLT2i), type 2 diabetes mellitus (T2DM), CAD, adverse event, GUTI

## Abstract

Sodium-glucose cotransporter 2 inhibitors (SGLT2i) were initially developed for the treatment of diabetes due to their antihyperglycemic activity. However, in the light of the most recent clinical studies, they are revolutionizing the approach to cardiovascular disease in patients with and without diabetes. We aimed to generate real-world data about the use of SGLT2i in patients with T2DM and coronary artery disease (CAD), focusing on their effectiveness in glycemic control, adherence, long-term efficacy, and safety outcomes. On the basis of the inclusion and exclusion criteria, 143 patients were enrolled. Patients were treated with canagliflozin (*n* = 33 patients; 23%), dapagliflozin (*n* = 52 patients, 36.4%), empagliflozin (*n* = 48 patients; 33.6%), or ertugliflozin (*n* = 10 patients; 7%) as monotherapy or in combination with other antidiabetic drugs. All patients performed a clinical visit, and their medical history, blood sampling, and anthropometric parameters were measured at discharge and at 1-year follow-up. The reduction in HbA1c % value at 12 months was significant (8.2 vs. 7.4; *p* < 0.001). Trends in body weight and body mass index also confirmed the positive effect of the treatment (*p* < 0.0001), as did the reduction in abdominal adiposity (expressed via waist circumference). At 1-year follow-up, 74.1% of patients were adherent to the treatment, and 81.1% were persistent to the treatment. A total of 27 patients (18.8%) had to discontinue treatment early due to drug intolerance caused by genitourinary infections (11.9%), the drub being permanently ineffective (HbA1c not at target or decreasing: 4.9%), or because of expressing. a desire not to continue (2%). No major drug-related adverse events (diabetic ketoacidosis, Fournier’s gangrene, lower-limb amputations) occurred at follow-up, while MACE events occurred in 14 patients (9.8%). In real-world patients with T2DM and CAD, SGLT2i have been effective in long-term glycemic control and the improvement in anthropometric indices with good tolerance, high adherence, persistence to treatment, and no major adverse events at 1-year follow-up.

## 1. Introduction

Sodium–glucose cotransporter 2 inhibitors (SGLT2i), which originally emerged as antihyperglycemic agents, block glucose resorption in the proximal tubule of the kidneys, and promote glucosuria, reducing blood glucose levels in an insulin-independent manner in Type 2 diabetes mellitus (T2DM) patients [1]. Their mechanism of action has recently shown beneficial effects on cardiovascular (CV) and renal biomarkers, such as blood pressure, body weight, and albuminuria [2], and several randomized clinical trials (RCTs) demonstrated the cardioprotective and nephroprotective effects of SGLT2i in diabetic patients [3,4,5,6].

On the basis of the results of CV outcome trials, the American Diabetes Association and the European Association for the Study of Diabetes (EASD) drafted recommendations for the treatment of patients with T2DM [7]. Accordingly, individuals with T2DM and established CV disease or with high CV risk indicators, including age ≥ 55 years with stenosis > 50% of the coronary, carotid, or lower extremity arteries, or with left ventricular hypertrophy should be treated with an SGLT2i or a glucagon-like peptide-1 receptor agonists (GLP1-RA) with demonstrated CV benefits regardless of HbA1c level or treatment with metformin. The 2019 guidelines of the European Society of Cardiology (ESC) in collaboration with the EASD [8] recommend treatment with SGLT2i as monotherapy in individuals with T2DM not yet on drug treatment, or in addition to metformin in cases of patients with established CV disease or those at high or very high risk defined on the basis of organ damage [8]. RCTs are the gold standard for the evaluation of therapy outcomes, providing the strongest scientific evidence. However, they are performed in a setting that is different from routine clinical practice; real-world settings offer complementary data and support clinicians in daily decision making.

Our study aims to generate real-world data about the use of these drugs by investigating their effectiveness and safety in a real-life setting. For this purpose, we report data on the use of SGLT2i by patients with T2DM and coronary artery disease (CAD) who had been treated with SGLT2i for diabetes, with an analysis of their effectiveness in glycemic control, adherence, long-term efficacy, and safety outcomes.

## 2. Results

### 2.1. Patient Characteristics

On the basis of the inclusion and exclusion criteria, 143 patients were considered in this analysis. The baseline clinical characteristics and medications of patients are presented in Table 1. The mean age of the cohort was 60.8 ± 11.2 years, and the majority (64.3%) were men. At the baseline, patients received aspirin at any dose (100%), P2Y12 inhibitors (100%), angiotensin-converting enzyme inhibitors, or angiotensin II receptor blockers (88.8%), b-blockers (91.6%), and any lipid-lowering therapy (100%). Regarding the index event, 39.9% of patients had been admitted for non-ST-elevation myocardial infarction (NSTEMI), 23.8% for ST elevation myocardial infarction (STEMI), and the remaining 36.4% for chronic coronary syndrome. Concerning the used SGLT2i and antidiabetic therapy, patients were treated with canagliflozin (*n* = 33 patients; 23%), dapagliflozin (*n* = 52 patients, 36.4%), empagliflozin (*n* = 48 patients; 33.6%), or ertugliflozin (*n* = 10 patients; 7%) as monotherapy or in combination with metformin (*n* = 104; 72.7%) and/or inhibitors of dipeptidyl peptidase 4 (DPP4-i) and/or glucagon-like peptide-1 receptor agonists (GLP-1 RA) (*n* = 5; 3.5%) and/or insulin (*n* = 70; 48.9%) and/or sulfonylureas (represented by gliclazide alone, *n* = 13; 9%).

### 2.2. Study End Points

A summary of the main changes at 1-year follow-up compared with the baseline with respect to. the biochemical, anthropometric, and clinical parameters is reported in Table 2.

The effect on the glucose metabolism was satisfactory, with a significant reduction in HbA1c % value at 12 months (8.2 vs. 7.4; *p* < 0.001) and with a good evolution of basal blood glucose, reduced by more than 30 mg/dL (175 vs. 143 mg/dL; *p* < 0.001). Trends in body weight and BMI also confirmed the positive effect of treatment (*p* < 0.0001), as did the reduction in abdominal adiposity (expressed via waist-circumference measurement). Systolic and diastolic blood pressure, total cholesterol and LDL-C, and renal function (expressed by creatinine) were not significantly changed.

Regarding adherence and persistence data, 74.1% of patients were adherent to treatment, and 81.1% were persistent to treatment at 1-year follow-up. Aside from the satisfactory therapeutic response in controlling diabetes, despite the careful educational process, as many as 27 patients (18.8%) had to discontinue treatment early due to drug intolerance caused by GUTIs (11.9%) on average after 6.5 months; the drug being permanently ineffective (HbA1c not at target or decreasing: 4.9%), on average after 7 months; after having expressed a desire not to continue (2%), on average after 6 months (Figure 1). In the present population, there were no reported adverse events (DKA, Fournier’s gangrene, lower limb amputations) or cases of bone fractures.

Regarding MACE, at 1-year follow-up, MACE events occurred in 14 patients (9.8%) (Figure 1). In our cohort, 9 patients (6.2%) experienced a new MI, 3 patients (2%) stroke, and 2 (1.4%) CV deaths occurred. In the multivariate logistic regression model, after adjusting for all known confounders, the onset as STEMI (OR = 1.51; 95%CI 1.01–1.94; *p* = 0.03) and the ejection fraction were independently associated with MACE (OR = 0.96; 95%CI 0.92–0.98; *p* < 0.001) (Table 3). Furthermore, the female sex and the presence of weakly managed diabetes were correlated with the occurrence of GUTIs (Table 3).

## 3. Discussion

The main findings of our real-life single-center retrospective observational study are the following. First, were highlighted the action and effectiveness of SGLT2i in positively influencing key metabolic and anthropometric parameters. Second, most patients demonstrate good tolerance for the drug by showing satisfactory adherence and persistence values. Third, with respect to the safety profile, a low UTI incidence rate was noted, and no major adverse events occurred. Fourth, the data with respect to MACEs are in line with published data to date, showing good cardiovascular protection.

In our cohort, at 1-year follow-up, patients showed a significant reduction in glycated hemoglobin, and most of them met the target of Hb1Ac < 7%. Furthermore, there was significant weight loss, and reductions in BMI and waist circumference with all the associated cardiovascular benefits. SGLT2i was studied as both monotherapy, and as add-on therapy to metformin, pioglitazone, metformin, and sulfonylurea, metformin and pioglitazone, metformin and DPP-4 inhibitors, and insulin (with or without oral hypoglycemic agents). Therapy with SGLT2i in T2DM2 reduces plasma glucose levels proportionally to the ambient glucose concentration and its glomerular filtration. There are no trials directly comparing individual SGLT2i. In network meta-analyses, placebo-corrected HbA1c reductions for monotherapy and dual therapy ranged from approximately 0.6 to 0.9% and 0.3 to 0.6%, respectively [9,10]. This finding is similar in magnitude to results with other oral hypoglycemic agents. In our population, 9% of patients were still treated with sulphonylureas. These drugs act by stimulating insulin secretion in a glucose-independent manner. They are a heterogeneous class of molecules with a different half-life and metabolism resulting in weight gain and accentuating the phenomenon of beta-cell depletion, leading to therapeutic failure more quickly than other drug classes [11]. They can cause hypoglycemia, and this peculiarity contraindicates their use in elderly patients in whom a hypoglycemic event could have significant and life-threatening consequences. From their registration studies, SGLT2i can be used as monotherapy combined with diet and exercise, but also with metformin and sulphonylureas, dipeptidyl peptidase 4 (DPP-4) inhibitors and glucagon-like peptide-1 (GLP-1) receptor agonists, and insulin. Limitations are mainly related to the reimbursement regulations. The use of sulphonylureas in combination with new drugs is limited to cost containment. Another notable finding with respect to drug therapy for diabetes is the low use of GLP1-RA (3.5% combined with DPP4). Despite the strong recommendations from national and international guidelines, only a minority of patients with T2DM are actually treated with hypoglycemic drugs with proven cardiovascular benefit [12]. A subanalysis of Italian patients in the CAPTURE study reported that about 15.8% of the patients were on GLP1-RA treatment, with no significant differences between the cohorts of patients with and without prior established cardiovascular disease. Real-world studies also confirmed that injectable GLP1-RAs are underused, and prescribed predominantly in patients with a long history of cardiovascular disease and with high mean HbA1c values (>8.2%) [13].

The benefit of the weight loss associated with SGLT2i therapy in diabetic patients is not to be underestimated. Most clinical trials with SGLT2i have demonstrated significant weight loss in the treated patients compared with that of the control or placebo-treated patients. A recent meta-analysis including most of the available studies on SGLT2i documented a weight loss of 1.74 kg vs. the placebo and 1.11 kg vs. the active treatment [2]. Since SGLT2 inhibition induces glycosuria, this weight loss could have been due to osmotic diuresis or to caloric loss, a logical consequence of glycosuria, calculable to be around 200–300 kcal per day. The measurements of fat mass by radiological techniques in patients treated with dapagliflozin or placebo associated with previous metformin treatment showed consistent reductions of 75% compared with 50% in the placebo group [14]. Both weight loss and fat mass reduction were associated with increased glycosuria, confirming the hypothesis that weight loss is attributable to urinary glucose loss. Studies also confirmed that dapagliflozin (in combination with sulfonylureas) and insulin could attenuate the well-known side effect of weight gain by these drugs [15,16,17].

Adherence and persistence data collected from our analysis show high adherence to treatment with SGLT2i, irrespective of the single active ingredient, and high persistence on treatment. In fact, 74.1% of the total showed MPR > 80%, and 81.1% were persistent at 1-year follow-up. Discontinuation was final in many subjects who had been advised to momentarily discontinue SGLT2i with a subsequent “rechallenge” (often with lower dosage than what was previously recommended). There are encouraging data from our analysis with respect to maintenance of therapy contrast, with two recent studies showing the opposite results. Hawley et al. [18], using Medicare claims data in patients ≥ 66 years old with T2DM, attempted to study adherence trajectories during the first year of SGLT2i therapy via the proportion of days covered (PDC). They found three adherence trajectory groups: low (23% of patients, mean PDC 17%), moderate (32%, mean PDC 50%), and high (45%, mean PDC 96%) adherence. A systematic review and random-effects meta-analysis of 22 studies and 123,854 individuals by Ofori-Asenso et al. [19] showed that PDC at 6 months and 1 year were 77% (95% confidence interval (CI) 0.72–0.82) and 72% (95% CI 0.66–0.77), respectively. The pooled proportion adherents (PDC ≥ 80%) at 6 months were 59.5% (95% CI 52.9–65.9), and 49.0% (95% CI 42.3–55.8) at 1 year. Persistence at 6 months, and 1 and 2 years were 80.1% (95% CI 75.8–84.0), 61.8% (95% CI 57.8–65.7), and 45.9% (95% CI 35.5–56.5), respectively. These data show poor real-world adherence and persistence to SGLT2i. In our experience, the discontinuation of the drug was often due to the onset of side effects and particularly urinary infections.

In the investigated patients in our cohort, the incidence of GUTIs seemed to be lower than what is commonly reported. This may have been due to the meticulous educational process on intimate hygiene that each patient had received at discharge. However, this is the most frequent cause of drug discontinuation. In a recent meta-analysis [20], no statistically significant differences were reported between SGLT2i and placebo users with regard to the occurrence of serious adverse events, while the increased risk of UTIs and GUTIs (events considered “nonserious”) was reiterated. In comparison with the placebo, SGLT2i were associated with a significantly increased risk of UTI. These increases in UTI risk were also present in comparison with other comparator drugs (such as metformin, sulfonylureas, and DPP4-i). The level of increased risk of GUTIs from SGLT2i in daily clinical practice is not well-understood, nor is it well-known whether this risk is dependent on sex or age, or whether it is more frequent at the onset of SGLT2i therapy. A retrospective cohort study [21] based on commercial U.S. data attempted to address these questions. In an initial analysis, the occurrence of genital candida infections (vaginitis and vulvovaginitis in women; balanitis, balanoposthitis, phimosis, and paraphimosis in men) in those taking SGLT2i vs. DPP4-i was evaluated. In a propensity-score-matched analysis of 129,994 women and 156,074 men, the hazard ratio was particularly higher for both women and men. This increased risk was similar when comparing SGLT2i vs. GLP1-RA users, with greater evidence for people ≥60 years old. No differences were found with respect to the type of SGLT2i used, but the increased risk is already evident in the first month of treatment and remains elevated throughout the period of therapy. The mechanism underlying the increased risk of GUTIs is still not well-known. It was initially proposed that glycosuria could be the trigger; however, in families with familial glycosuria, there is no reported increase in the frequency of GUTIs [22]. It was also hypothesized that this could be due to the hyperglycemia, but subanalyses were performed to link the degree of metabolic control (according to HbA1c) with the higher incidence of infections did not show a positive association between higher HbA1c and higher incidence or severity of GUTIs [23]. In the healthy urinary tract, the epithelium of the urethra, bladder, ureters, and collecting ducts express and secrete a cocktail of antimicrobial peptides and proteins (AMPs) that effectively restrict bacterial proliferation. The expression of many of these AMPs depends on the classical insulin signaling pathway. In the direction of urine flow along the collecting ducts, the concentration of the antimicrobial cocktail increases, creating an increasingly effective immune barrier [23]. However, in the context of insulin resistance, the expression of insulin-dependent AMPs is suppressed, creating a less optimal antimicrobial environment, allowing for viable microbes to ascend into the collecting ducts (Figure 2). In this scenario, the glycosuric mechanism of SGLT2i could create a favorable environment for the growth of pathogenic bacterial flora.

Our data show a MACE rate of 9.8% at 1-year follow-up. These were mainly entrained by recurrences of MI reflecting, but a multiplicity of concurrences in optimizing secondary prevention therapy may underlie recurrences. Empagliflozin, Cardiovascular Outcomes, and Mortality in Type 2 Diabetes (EMPA-REG OUTCOME) was the first of studies evaluating the safety and effectiveness of an SGLT2i. Patients with T2DM (N = 7028) and established CV disease (MI, CAD, unstable angina, cerebral vascular accident, or peripheral arterial disease) were randomly assigned to receive 10 mg empagliflozin or 25 mg placebo once daily for a 3-year follow-up period [6]. Empagliflozin, added to standard antidiabetic therapy, reduced the composite risk of cardiovascular death, nonfatal MI, and nonfatal stroke by 14% (hazard ratio 0.86; CI 95.02%, 0.74–0.99; *p* = 0.04 for superiority), mainly at the expense of a reduction in cardiovascular death of 38%. It also reduced all-cause death by 32% and hospitalization for heart failure by 35% compared to the placebo. The two dosages of empagliflozin showed similar hazard ratios for cardiovascular outcomes. More recently, following the trail of this amazing trial, the CANVAS Program [4] and the DECLARE-TIMI 58 trial [5] confirmed the lower rate in cardiovascular death or hospitalization for heart failure in SGLT2i groups. However, in contrast with the EMPA-REG OUTCOME trial result, they failed to show the positive effect of canagliflozin 100 mg/300 mg and dapagliflozin 10 mg on reducing the rate of all-cause death. The main differences in the results of these trials were due to the characteristics of the study populations. In the EMPA-REG study, 99.5% of the patients had a prior cardiovascular disease, whereas in the CANVAS Program, only 66% of the patients had a prior cardiovascular disease, and the remaining 34% were in the high-cardiovascular-risk category. In the Evaluation of Ertugliflozin Efficacy and Safety Cardiovascular Outcomes Trial (VERTIS CV), ertugliflozin was noninferior to placebo with respect to MACE [3].

All these described benefits have their basis at the molecular level, some known and well-studied, and others are part of ongoing, indepth investigations [24]. Cardiac muscle tissue uses glucose and fatty acids as energy sources. Glucose, when oxidized, generates fewer ATP molecules and requires more oxygen for its oxidation. The SGLT2i creates a change in the cardiac metabolic source because, by reducing glucose levels, the cardiomyocyte uses fatty acids as a source of energy (less oxygen requirement and more energy), leading to increased lipolysis and release of non-esterified fatty acids [25]. This phenomenon induced by SGLT2i decreases body fat and reduces steatosis, which is beneficial at the cardiovascular level [26]. To this we can add the reduction in sympathetic nervous system activity and increased oxygen supply through the renal stimulation of erythropoietin. SGLT2i is emerging as a basic pillar in the treatment of patients with cardiovascular disease, and, due to published studies in recent years, there are enough data to confidently estimate its efficacy in reducing morbidity and mortality, and the number of hospitalizations and complications of cardiovascular disease.

In our cohort, heart-failure patients with reduced ejection fraction comprised 30.7%. In these patients, it is of primary importance to institute therapy as early as possible with SGLT2i. In the course of clinical trials aimed at evaluating their cardiovascular safety in diabetic patients, SGLT2i, particularly dapagliflozin and empagliflozin, were safe, and their use was associated with a better prognosis in terms of reducing both the risk of developing heart failure and adverse events in those already diagnosed with heart failure. On the basis of this evidence, RCTs were conducted to evaluate the effect of SGLT2i in patients with heart failure regardless of the presence of diabetes mellitus. The new and intriguing recommendation results from two RCTs, in particular, the Dapagliflozin and Prevention of Adverse Outcomes in Heart Failure (DAPA-HF) trial [27], and the empagliflozin outcome trial in patients with chronic heart failure and a reduced ejection fraction (EMPEROR-Reduced) trial [28], which demonstrated significantly reduced cardiovascular death and hospitalization for heart failure with dapagliflozin and empagliflozin, respectively, regardless of the presence or absence of diabetes. The 2021 ESC guidelines [29] and the most recent 2022 American Heart Association/American College of Cardiology/Heart Failure Society of America (AHA/ACC/HFSA) guidelines [30] for the management of heart failure recommend a paradigm shift in the treatment of heart failure with reduced ejection fraction patients by introducing SGLT2i as the fourth therapeutic pillar, in addition to optimal medical therapy with angiotensin-converting enzyme inhibitor or angiotensin receptor blocker or angiotensin receptor–neprilysin inhibitor, beta-blockers, and mineralocorticoid receptor antagonists.

A point of discussion in the management of these patients concerns the prescription and reimbursability of this class of drugs. During the observation period, patients were identified and referred to a diabetes specialist who was the only one authorized to prescribe in Italy. This involved close interdisciplinary collaboration to provide the patient with the best care. In 2022, the Italian Drug Agency issued Note 100, which regulates the reimbursability of these drugs, even by the cardiologist.

The strengths of this observational study concern the tolerance of the drug in clinical practice at an outpatient clinic dedicated to these patients, which also allows for the optimal management of complications, and real or presumed adverse effects. In this regard, a lower drug discontinuation rate was experienced than that in trials, managing even adverse effects such as GUTIs while trying not to lose the cardiovascular benefit of this class of drugs in patients with T2DM and CAD.

### Study Limitations

This study has several limitations. The observational design of the study limited our conclusions. The sample size was not representative of the whole population because the data came from a single center. In addition, an observational study may contain inaccuracies, such as the completeness of the data or coding that could result in biases. Further detailed data on heart failure condition are described through NT-proBNP levels or echocardiographic data. Moreover, the risk of potential concealed conditions must be taken into account.

## 4. Materials and Methods

### 4.1. Study Design and Study Population

This prospective observational single-center cohort study was conducted according to the Strengthening the Reporting of Observational Studies in Epidemiology (STROBE) Statement. We enrolled patients admitted to the cardiology department of Sant’Anna e San Sebastiano Hospital (Caserta, Italy). The study was conducted according to the institutional standards, national legal requirements, and the Declaration of Helsinki. Informed consent was obtained from all patients. Patients were included if they (i) had been admitted to the department of cardiology with a diagnosis of CAD, (ii) had had a diagnosis of T2DM at the time of admission and had been introduced SGLT2i (canagliflozin, empagliflozin, dapagliflozin, ertugliflozin) during hospitalization or at discharge (according to eligibility criteria for SGLT2i reimbursement of the Italian Drug Agency (AIFA)), (iii) had been on metformin therapy (unless intolerant or with contraindications), and (iv) had exhibited inadequate glycemic control (glycated hemoglobin HbA1c greater than 53 mmol/mol or 7.0%). This class of drugs were precribed by the diabetes specialist.

We applied the following exclusion criteria: (i) patients with a diagnosis of Type 1 diabetes mellitus; (ii) chronic cystitis and/or recurrent urinary tract infections; (iii) therapy with SGLT2i within 8 weeks prior to hospitalization; (iv) uncontrolled hyperglycemia with a glucose level > 240 mg/dL (>13.3 mmol/L) after an overnight fast; (v) incomplete clinical data or declining participation in the study. We collected demographic and clinical information, laboratory results, and drug therapy data at the baseline from each patient’s electronic medical records from January 2020 to December 2021. Clinical outcomes were followed up for 12 months; outcome data were acquired through telephonic or in-person follow-ups, and through collaborations with the patients’ general practitioners and diabetologists.

### 4.2. Follow-Up End Points

Treatment efficacy for glycemic control was evaluated by comparing HbA1c levels at discharge and at 1-year follow-up. Information was compared related to weight, anthropometric indices (body mass index (BMI), waist circumference), arterial blood pressure, and renal function at discharge and at 1-year follow-up. In addition, adherence and persistence to treatment with SGLT2i at 12 months were evaluated. The safety end points were genitourinary tract infections (GUTIs), episodes of diabetic ketoacidosis (DKA), and any other adverse event attributable to the administration of the drug.

At 1 year, descriptive outcome analysis of the efficacy end points was performed, represented by a composite endpoint ((3-point major adverse cardiovascular events (MACE)): CV death (including fatal stroke and fatal myocardial infarction (MI)), nonfatal MI, and nonfatal stroke.

Adherence was assessed via the medical possession rate (MPR), defined as the ratio in percentage between the number of days of therapy, and the interval between the first and last prescriptions. An MPR ≥ 80% indicated high adherence, and <40% indicated low adherence. Persistence to treatment was defined as the time between initiation and discontinuation of the prescribed drug treatment (discontinuation if we had observed a greater time gap than 60 days, defined as the grace period).

GUTIs were classified into (a) urinary tract infections (UTIs) including cystitis, pyelonephritis, and prostatitis, and (b) genital tract infections including urethritis, cervicitis, epididymitis, genital ulcerative diseases, endometritis, and pelvic inflammatory disease. Infection was inferred with any symptom indicating a UTI: dysuria, fever, increased urinary frequency, urinary incontinence, suprapubic pain, or hematuria.

MI was defined according to the universal MI definition by Thygesen et al. [31]. Stroke was defined as an acute episode of neurologic dysfunction attributed to a central-nervous-system vascular cause. Stroke should be documented by imaging (e.g., computed tomography scan or magnetic resonance imaging). Cardiovascular death was defined as a death involving cardiac arrest, often with suggestive symptoms of myocardial ischemia and accompanied by new ST elevation, new left bundle branch block, and/or evidence of thrombus in coronary angiography and/or autopsy.

### 4.3. Statistical Analysis

Categorical variables were the frequencies and percentages, and continuous ones were the mean and SD. The Kolmogorov–Smirnov test was applied to check the normality. To evaluate the significance of differences between groups, the χ^2^ test was used, or Fisher’s exact test whenever the samples were rather small. According to the sample size and the normality distribution, Student’s *t* test (assumed to be equal or unequal variances as appropriate) was used to compare continuous variables. We produced multivariate logistic regression and multinomial logistic regression models to assess the predictability of variables on the occurrence of composite endpoints, and each component is presented as the odds ratio (OR) and 95% confidence intervals (CI), using *p* < 0.1 in the univariate analyses for inclusion. The significance threshold was set at *p* = 0.05 (2-tailed). All analyses were performed using SPSS for Windows (version 25; SPSS, Inc., Chicago, IL, USA) and R software (CRAN^®^ 3.3.4).

## 5. Conclusions

In real-world patients with previous T2DM and CAD, SGLT2i have been effective in long-term glycemic control and the improvement of anthropometric indices. Despite the presence of some adverse events, this class of drugs seemed to be well-tolerated in our cohort, with high adherence and persistence to treatment, and no major adverse events at 1-year follow-up. The appropriate identification of T2DM patients with CAD for treatment with SGLT2i according to what is suggested by international guidelines could ensure long-term benefits with regard to hard CV endpoints.

## Figures and Tables

**Figure 1 ijms-24-08099-f001:**
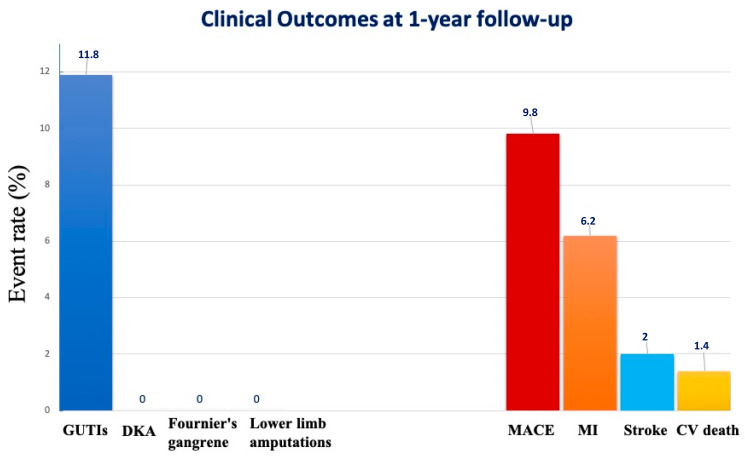
Clinical outcomes at 1-year follow-up in patients with T2DM and CAD treated with SGLT2i. CV: cardiovascular. DKA: diabetic ketoacidosis. GUTIs: genitourinary tract infections. MACE: major adverse cardiovascular events. MI: myocardial infarction.

**Figure 2 ijms-24-08099-f002:**
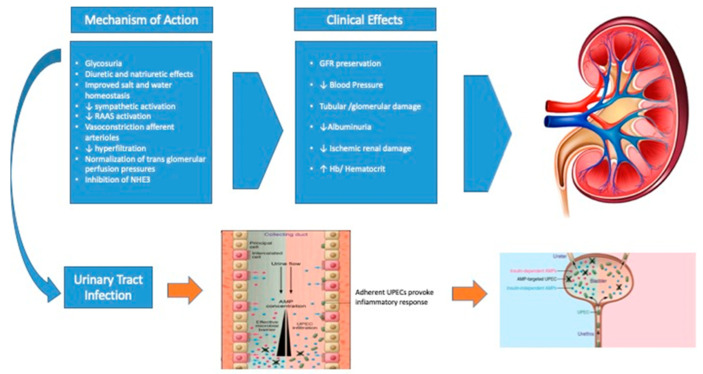
Biological and molecular mechanisms of SGTL2i and pathophysiological mechanism of genitourinary tract infections. AMP: antimicrobial peptides and proteins. GFR: glomerular filtration rate. NHE3: Na+/H+ exchanger isoform 3. RAAS: renin-angiotensin-aldosterone system. UPEC: Uropathogenic Escherichia coli.

**Table 1 ijms-24-08099-t001:** Baseline characteristics and concomitant medications in T2DM patients receiving SGLT2i.

Baseline Characteristics	T2DM Patients Treated with SGLT2i 143 pts.
Age—years (mean ± SD)	60.8 ± 11.2
Female sex—no. (%)	51 (35.6)
Current Smoker—no. (%)	33 (23.1)
Hypertension—no. (%)	115 (80.4)
Diabetes—no. (%)	143 (100)
Hypercholesterolemia—no. (%)	104 (72.7)
>1 Prior MI—no. (%)	35 (24.5)
LVEF % (±SD)	48 (±8)
HFrEF—no. (%)	44 (30.8)
Type of index event (CAD)—no. (%)	
CCS	52 (36.4)
STEMI	34 (23.8)
NSTEMI/UA	57 (39.9)
Concomitant Medication at enrollment—no. (%)	
Aspirin at any dose—no. (%)	143 (100)
P2Y12 inhibitors—no. (%)	143 (100)
Clopidogrel—no. (%)	56 (39.1)
Ticagrelor—no. (%)	80 (56.7)
Prasugrel—no. (%)	7 (4.9)
Statin— no. (%)	140 (97.9)
Ezetimibe—no. (%)	73 (51)
PCSK9i—no. (%)	14 (9.7)
Beta-blockers— no. (%)	131 (91.6)
ACE inhibitors or ARB— no. (%)	127 (88.8)
ARNI—no. (%)	35 (24.5)
SGLT2 inhibitors—no. (%)	143 (100)
Canagliflozin—no. (%)	33 (23)
Dapagliflozin—no. (%)	52 (36.4)
Empagliflozin—no. (%)	48 (33.6)
Ertugliflozin—no. (%)	10 (7)
Antidiabetic therapy	
Metformin—no. (%)	104 (72.7)
DPP4-I and/or GLP-1 RA—no. (%)	5 (3.5)
Insulin—no. (%)	70 (48.9)
Sulfonylureas—no. (%)	13 (9)

ACE: angiotensin-converting enzyme; ARB: angiotensin II receptor antagonists; bid: bis in die; DPP4-i: Inhibitors of dipeptidyl peptidase 4; GLP-1 RA: glucagon-like peptide-1 receptor agonists; MI: myocardial infarction; NSTEMI: non ST-elevation myocardial infarction; SGLT2i: Sodium-glucose cotransporter 2 inhibitors; STEMI: ST-elevation myocardial infarction, SD: standard deviation, T2DM: type 2 diabetes mellitus; UA: unstable angina.

**Table 2 ijms-24-08099-t002:** Variation between discharge visit and 1-year follow-up for metabolic, biochemical, anthropometric, and clinical parameters.

	Discharge	1-Year Follow-Up	*p*-Value
HbA1c (%) ± SD	8.2 ± 1.7	7.4 ± 1.4	<0.001
Basal Blood Glucose (mg/dl) ± SD	175 ± 61.4	143 ± 40.3	<0.001
Weight (kg) ± SD	92.1 ± 17.6	89.5 ± 17.1	<0.001
Body Mass Index ± SD	32.8 ± 6.2	30.9 ± 6.4	<0.001
Waist Circumference (cm) ± SD	110.8 ± 13.8	108.7 ± 13.4	0.02
Systolic blood pressure (mmHg) ± SD	134.8± 14.2	135.2 ± 18.1	0.6
Diastolic blood pressure (mmHg) ± SD	78.8 ± 10.1	78.5 ± 9.2	0.81
Total cholesterol (mg/dl) ± SD	177.8 ± 37.3	178.8 ± 36.2	0.45
LDL-C (mg/dl) ± SD	89.8 ± 25.3	85.6 ± 26.4	0.07
Creatinine (mg/dl) ± SD	0.92 ± 0.5	0.9 ± 0.3	0.49
Creatinine Clearance (ml/min/1.73 m^2^)	65.4 ± 11.1	63.3 ± 10.5	0.1

LDL: low-density lipoprotein. SD: standard deviation.

**Table 3 ijms-24-08099-t003:** Multivariable analysis: predictors of MACE and GUTIs.

MACE			GUTIs		
Variables	OR [CI, 95%]	*p*-Value	Variables	OR [CI, 95%]	*p*-Value
Age	1.07 [0.99–1.13]	0.434	Age	1.06 [0.98–1.09]	0.357
Gender, Male	0.82 [0.43–1.56]	0.537	Gender, Male	0.92 [0.43–0.99]	0.04
STEMI	1.51 [1.01–1.94]	0.03	HbA1c > 8%	1.10 [1.01–1.91]	0.03
LVEF	0.96 [0.92–0.98]	<0.001	BMI	1.04 [0.90–1.97]	0.09
Complete revascularization	0.72 [0.39–1.34]	0.302	Creatinine	0.75 [0.38–1.48]	0.401

BMI: body mass index; GUTIs: genitourinary tract infections; LVEF = left ventricular ejection fraction; STEMI = ST elevation myocardial infarction; VT = ventricular tachycardia.

## Data Availability

The data presented in this study are available on request from the corresponding author.

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
