# Peer review of "Sodium–Glucose Cotransporter 2 Inhibitors in Patients with Diabetes and Coronary Artery Disease: Translating the Benefits of the Molecular Mechanisms of Gliflozins into Clinical Practice"

_ijms, 2023, doi:10.3390/ijms24098099_

Round 1

Reviewer 1 Report

Dear authors

thank you for your research activity. The paper is interesting and well written. Please better explain strengths and limitations of the study and specify if SGLT2i were prescribed directly by cardiologists or by diabetologists: this is a key point for the importance of this drug category not only in terms of diabetes but in particular in terms of cardiovascular prevention. 
Moreover please revise English form all over the text (minor spell check required)

Author Response

We thank the Reviewer for allow us to add these points to the discussion. Strengths and limitations were integrated into the discussion. In addition, it was specified which specialist prescribed SGLT2i.  For this purpose, the text was amended as follows:

“A point of discussion in the management of these patients concerns the prescribing and reimbursability of this class of drugs. During the observation period, patients were identified and referred to the diabetes specialist who was the only one authorized to prescribe in Italy. This involved close interdisciplinary collaboration to provide the pa-tient with the best of care. Fortunately, as of 2022, the Italian Drug Agency has issued "Note 100," which regulates the reimbursability of these drugs even by the cardiologist.

The strengths of this observational study concern the tolerance of the drug in clinical practice at an outpatient clinic dedicated to these patients, which also allows optimal management of complications, and adverse effects, real or presumed. In this regard, a lower drug discontinuation rate was experienced than in trials, managing even adverse effects such as GUTIs while trying not to lose the cardiovascular benefit of this class of drugs in patients with T2DM and CAD.”

(page 6, lines 51-54; page 7, lines 1-8)

Reviewer 2 Report

Cesaro and colleagues in this original research article entitled “Sodium-glucose cotransporter 2 inhibitors in patients with diabetes and coronary artery disease: translating the benefits of the molecular mechanisms of gliflozins into clinical practice” report real-world data about the use of SGLT2i in patients with T2DM and CAD at their Institution.

The study included 143 patients treated with different SGLT2i molecules (canagliflozin, dapagliflozin, empagliflozin, or ertugliflozin) in order to describe the class effect, instead of single glyphozine molecules, in this very-high-risk population.

The authors investigated the effectiveness of SGLT2i on conventional targets including HbA1c, body weight, and waist circumference, reporting a significant reduction since the beginning of therapy. However, the principal value of the article was the assessment of medical adherence to SGLT2i in this real-world population. Medical adherence was assessed by medical possession rate (MPR) and reasons of discontinuation were systematically reported. They reported that 74.1% of patients were adherent and 81.1% were persistent in treatment up to one year. The main reason for discontinuation was genito-urinary infections.

This study deals with a current topic of considerable clinical interest by providing a picture of SGLT2i use in the real world.

The manuscript is well-written; the tables are descriptive of the characteristics of the study population. I really appreciated figures that are clear and impactful to the reader.

I have just a few minor comments for the authors:

- Abstract: “No major adverse events occurred at follow-up, while MACE events occurred in 14 patients (9.8%)”. If I understand well when the authors said that no major adverse events occurred they refer to DM-related events (DKA, Fournier’s gangrene, lower limb amputations) or adverse events attributable to the administration of the drug. I would suggest clarifying this point in the text.

- In the study limitations, I would suggest including the absence of information on NT-proBNP and transthoracic echocardiography. 

Author Response

We thank the Reviewer for her/his comment. We specified in the abstract the absence of major drug-related adverse events as suggested by the reviewer.

Reviewer 3 Report

This distinguished group of investigators aimed at exploring real-world data about the use of SGLT2i in patients with T2DM and coronary artery disease (CAD) focusing on their effectiveness in glycemic control, adherence, long-term efficacy, and safety outcomes. In 143 T2DM + CAD patients, Authors found that SGLT2i have been shown to be effective in long-term glycemic control and improvement of anthropometric indices, with good tolerance, high adherence, and persistence to treatment. No major adverse events were observed at 1-year follow-up.

The article is interesting, well-conducted. Discussion section is nicely written.

Some major issues should be acknowledged:

- I would suggest the following order for paragraphs: introduction, methods, results, discussion, conclusions.

- please comment on sulphanylureas use in 9% of the study population

- please comment on lower rate of patients on GLP1ra therapy (3.5% combined to DPP4)

- mean LDL-cholesterol values seems not being on target despite a modest adoption of PCSK9i therapy

- I would suggest multivariate analysis/cox regression analysis to elucidate which are the main covariates associated to MACE and/or per each outcome variable (MI, stroke, death) or GUTIs.

- creatinine clearance should be added to Table 1 if available to better describe study population

- If available, please add inflammatory markers to Table 1 and to multivariate analysis/cox regression analysis

Author Response

The article is interesting, well-conducted. Discussion section is nicely written.

We are thankful to the Reviewer for the constructive comments.

Some major issues should be acknowledged:

- I would suggest the following order for paragraphs: introduction, methods, results, discussion, conclusions.

We are thankful to the Reviewer for the comment. Dear reviewer, In our submission the order of the paragraphs was as you suggest. However, the journal's editorial rules provide for the order that you find instead.

For more details, this part can be found on the journal's website: https://www.mdpi.com/journal/ijms/instructions       

- please comment on sulphanylureas use in 9% of the study population

According to the Reviewer’s request, we have briefly discussed the use of sulphonylureas.

“In our population, 9% of patients were still treated with sulphonylureas. This drugs act by stimulating insulin secretion in a glucose-independent manner. They are a heterogeneous class of molecules with different half-life and metabolism; result in weight gain and accentuate the phenomenon of beta-cell depletion, leading more quickly to therapeutic failure than other drug classes. They can cause hypoglycemia, and this peculiarity contraindicates their use in the elderly patient, in whom a hypoglycemic event can have significant and life-threatening consequences. From their registration studies, SGLT2i can be used either as monotherapy combined with diet and exercise, but also with metformin and sulphonylureas, DPP-4 (dipeptidyl peptidase 4) inhibitors and GLP-1 (glucagon-like peptide-1) receptor agonists, and insulin. Limitations are mainly related to the reimbursement regulations. The use of sulphonylureas in combination with new drugs is limited to the hypothesis of cost containment.” (page 4, lines 8-19).  

- please comment on lower rate of patients on GLP1ra therapy (3.5% combined to DPP4)

Thank you because this comment allows us to raise a crucial point for the discussion on the contemporary treatment of patients with T2DM and CAD and allows us to shine a spotlight on the underutilization of drugs with proven cardiovascular benefit such as GLP1-RAs. To discuss this we added in the text:

“Another notable finding with respect to drug therapy for diabetes is the low use of GLP1-RA (3.5% combined with DPP4). Despite strong recommendations from national and international guidelines, only a minority of patients with T2DM to date are actually treated with hypoglycemic drugs with proven cardiovascular benefit. The subanalysis of Italian patients in the CAPTURE study reports that about 15.8% of patients were on GLP1-RA treatment, with no significant differences between the cohorts of patients with or without prior established cardiovascular disease. Real-world studies also confirm that to date, injectable GLP1-RAs are underused and prescribed predominantly in patients with a long history of cardiovascular disease and with high mean HbA1c values (>8.2%).” (page 4, lines 20-29).  

- mean LDL-cholesterol values seems not being on target despite a modest adoption of PCSK9i therapy

The point raised fully captures the lack of target attainment in secondary prevention patients. At an analysis of adherence to lipid-lowering therapy at 1 year, it was found that patients had partly abandoned statin therapy or there had been inappropriate dose de-escalation. PCSK9 inhibitors are used in 9.7 percent of cases, but at 1-year follow-up in addition to recommending oral therapy, PCSK9i therapy was implemented.

- I would suggest multivariate analysis/cox regression analysis to elucidate which are the main covariates associated to MACE and/or per each outcome variable (MI, stroke, death) or GUTIs.

Although there are few events at follow-up and it is only a descriptive analysis, following the reviewer's recommendations, we performed a multivariate analysis to assess which variables were most associated with MACEs or GUTIs.

MACE

GUTIs

Variables

OR [CI, 95%]

p-value

Variables

OR [CI, 95%]

p-value

Age

1.07 [0.99-1.13]

0.434

Age

1.06 [0.98-1.09]

0.357

Gender, Male

0.82 [0.43-1.56]

0.537

Gender, Male

0.92 [0.43-0.99]

0.04

STEMI

1.51 [1.01-1.94]

0.03

HbA1c >8%

1.10 [1.01-1.91]

0.03

LVEF

0.96 [0.92-0.98]

<0.001

BMI

1.04 [0.90-1.97]

0.09

Complete revascularization

0.72 [0.39-1.34]

0.302

Creatinine

0.75 [0.38-1.48]

0.401

- creatinine clearance should be added to Table 1 if available to better describe study population

Thank you for the comment. It allows us to better describe the population. Information on creatinine clearance has been added to Table 2, at baseline and 1-year follow-up.

- If available, please add inflammatory markers to Table 1 and to multivariate analysis/cox regression analysis.

Thank you for the comment. Unfortunately, there are no data on inflammatory markers.

- In the study limitations, I would suggest including the absence of information on NT-proBNP and transthoracic echocardiography.

As suggested by the Reviewer, we added in the limitations the absence of more detailed echocardiographic data and data on NT-proBNP: “It should be pointed out that further detailed data on heart failure condition described through NT-proBNP levels or echocardiographic data.”

Reviewer 4 Report

Four Sodium–Glucose Cotransporter 2 Inhibitors (SGLT2is), i.e., empagliflozin, dapagliflozin, canagliflozin, and ertugliflozin were approved by the European Medicines Agency and the US Food and Drug Administration on the basis of large-scale randomized controlled trials and documented cardiovascular benefits. The cardiovascular benefits of SGLT2is are beyond glycemic control and include cardioprotective effects.

The article is devoted to the assesment of effectiveness of SGLT2i in glycemic control, adherence, long-term efficacy, and safety outcomes among patients with type 2 diabetis and coronary artery disease (CAD).

The strengths of this article: it was added some up-to-date information in the population of patients with diabetes and CAD, especially in those treated with SGLT2is.

in my opinion, the manuscript might become more useful, if the information about the obesity and chronic heart failure will be added.

Author Response

Information regarding the presence of HFrEF was added (Table 1) and the point raised was discussed in the text:

In our cohort, heart failure patients with reduced ejection fraction were 30.7%. In these patients, it is of primary importance to institute therapy as early as possible with SGLT2i. In the course of clinical trials aimed at evaluating their cardiovascular safety in diabetic patients, SGLT2i, particularly dapagliflozin and empagliflozin, have not only been shown to be safe; their use has been found to be associated with a better prognosis both in terms of reducing the risk of developing heart failure and in terms of reducing adverse events in those already diagnosed with heart failure. Based on this evidence, RCTs were conducted aimed at evaluating the effect of SGLT2i in patients with heart failure re-gardless of the presence of diabetes mellitus. The new and intriguing recommendation results from two RCTs, in particular the Dapagliflozin And Prevention of Adverse out-comes in Heart Failure (DAPA-HF) trial [27] and the Empagliflozin Outcome Trial in Patients with Chronic Heart Failure and a Reduced Ejection Fraction (EMPER-OR-Reduced) trial [28], which demonstrated significantly reduced cardiovascular death and hospitalization for heart failure respectively with dapagliflozin and empagliflozin, regardless of the presence or absence of diabetes. The 2021 ESC guidelines [29] and the most recent 2022 American Heart Association/American College of Cardiology/Heart Failure Society of America (AHA/ACC/HFSA) guidelines [30] for the management of heart failure recommend a paradigm shift in the treatment of heart failure with reduced ejection fraction patients by introducing SGLT2i as the fourth therapeutic pillar in addition to optimal medical therapy with angiotensin-converting enzyme inhibitor or angiotensin receptor blocker or angiotensin receptor–neprilysin inhibitor, beta-blockers, and min-eralocorticoid receptor antagonists.” (page 6, lines 31-52).

Round 2

Reviewer 3 Report

Authors fully replied to Reviewers' suggestions. Manuscript has been markedly improved.